# Experiences of seeking healthcare across the border: lessons to inform upstream policies and system developments on cross-border health in East Africa

Freddie Ssengooba,[1,2] Doreen Tuhebwe [ID],[3] Steven Ssendagire,[3] Susan Babirye,[3] Martha Akulume,[3] Aloysius Ssennyonjo [ID],[1,2] Arthur Rutaroh,[4] Leon Mutesa,[5] Mabel Nangami[6]

For numbered affiliations see end of article.

**Correspondence to**
Dr Freddie Ssengooba;
sengooba@musph.ac.ug

## ABSTRACT

**Objectives** This study explored the experiences of accessing care across the border in East Africa.

**Participants** From February to June 2018, a cross-sectional study using qualitative and quantitative methods was conducted among 279 household adults residing along selected national border sites of Uganda, Kenya and Rwanda and had accessed care from the opposite side of the border 5 years prior to this study.

**Setting** Access to HIV treatment, maternal delivery and childhood immunisation services was explored. We applied the health access framework and an appreciative inquiry approach to identify factors that enabled access to the services.

**Measures** Exploratory factor analysis and linear regression were used for quantitative data, while deductive content analysis was done for the qualitative data on respondent's experiences navigating health access barriers.

**Results** The majority of respondents (83.9%; 234/279) had accessed care from public health facilities. Nearly one-third (77/279) had sought care across the border more than a year ago and 22.9% (64/279) less than a month ago. From the linear regression, the main predictor for ease of access for healthcare were "ease of border crossing' (regression coefficient (RegCoef) 0.381); 'services being free' (RegCoef 0.478); 'services and medicines availability' (RegCoef 0.274) and 'acceptable quality of services' (RegCoef 0.364). The key facilitators for successful navigation of access barriers were related to the presence of informal routes, speaking a similar language and the ability to pay for the services.

**Conclusion** Communities resident near national borders were able to cross borders to seek healthcare. There is need for a policy environment to enable East Africa invest better and realise synergies for these communities. This will advance Universal Health Coverage goals for communities along the border who represent the far fang areas of the health system with multiple barriers to healthcare access.

## INTRODUCTION

Border resident communities are the population settlements, rural or urban, in close proximity to the borders.[1] This term usually applies to international borders and the border resident communities of the two neighbouring countries are usually economically and socially interdependent[2] and have more in common with their sister communities across the border than with communities on their own side of the border. Residents of both countries cross the border for a variety of reasons, including work, shopping, healthcare, visits with friends or relatives, and others.[3]

Increasingly, border resident communities cross to access care in each other's country.[4] This is in partly due to challenges of accessing the needed care in the home country where the nearest health facility may be far away into their country and the border residents are located at the periphery with physical and economic access barriers. Cross-border healthcare can also enhance individual choice, by providing added option to obtain healthcare which may either exhibit a better quality or be available at a lower cost in comparison with their national provision system.[5][6] A study by Su *et al* showed that one of the most significant predictors of healthcare utilisation in Mexico by US citizens were

lack of US health insurance coverage and dissatisfaction with the quality of care in the USA.[7] Scholars have also proposed several frameworks that can be used to understand facilitators and barriers to health access.[8–10]

**STRENGTHS AND LIMITATIONS OF THIS STUDY**

⇒ The study involved the use of both qualitative and quantitative data which strengthened the triangulation of results.
⇒ Having multicountry study sites in the East Africa Community strengthened the study.
⇒ This paper only presents the views of respondents that successfully navigated access barriers.
⇒ We did not include communities along the numerous informal cross points which may have different contexts and experiences.

However, access to care in the neighbouring country may not always be easy given the several barriers experienced along the pathway to care.[2] Some of the barriers to physical access may include challenges of poor road network and terrain and lack of identity documentation.[11] It could be an issue of financial barriers like high cost of care across the border despite of physical proximity, need for foreign currency and untransferability of health insurance policy to a neighbouring country.[5] Other hindrances include; lack of quality healthcare infrastructure, healthcare workforce shortages and language barriers that pose acceptability challenges.[3 12] These barriers to access and utilisation are often times rooted in the differences within legal and institutional frameworks that confer entitlement to healthcare between countries as well as creating differences in healthcare protocols.[13]

Although states have obligations to ensure their citizens have access to all health services,[14] this is sometimes not fulfilled especially for communities residing along international borders.[15] Because the nearest health facility or health services needed may be across the border, individuals and communities usually navigate these barriers in order to access care in the neighbouring country[16] though not institutionally recognised or planned for by the country receiving the clients. In the context of the East African Community (EAC), member states (Uganda, Kenya, Tanzania, Rwanda, Burundi and South Sudan) have a progressive plan to increase the interaction and freedom of movements of commercial goods and people.

Literature has shown several mechanisms that border residents use to navigate health access barriers. A few of the strategies include using social and kinship networks which facilitate cross-border exchange of information about where care is available and where quality can be accessed.[6 11] This literature also shows that people take advantage of the cultural context and the ethnolinguistic similarities to navigate barriers such as ineligibility or special preconditions for non-citizens to healthcare services.[17] Because some of the border residents live as one community in the two opposite countries, there are usually many informal crossings through non-official crossing points for trade and inherently promote cross-border health access.

Border communities, regardless of their size, are often regarded as peripheral in terms of social programmes but paradoxically have high priority in terms of national security and in allocation of citizenry privileges.[18] Of recent, border communities have also become of importance for public health surveillance due to threats like multidrug resistant TB,[19] Ebola outbreak experience in West Africa[20] and recently, the novel COVID-19. These threats can and have traversed borders as a result of inflows and outflows of people and goods across borders. The disease surveillance objectives have in some places forced health systems across the borders to cooperate in the control of disease outbreaks.[21] It is not clear how this cooperation is extending to support day-to-day healthcare needs of communities along the borders. As another paper that we are working on demonstrates, there are major elements of state obligations in ensuring access to health for those in need since 'health is a right'. However, these obligations are limited to citizens apart from a few aspects like epidemic control events and the high end chronic care centres of excellence that are part of the regional efforts to address cross-border health access.

Leaving no one behind is the mantra for Universal Health Coverage (UHC) and the Sustainable Development Goals.[22] Communities like those residing along the EAC borders must be targeted because they represent the far fang areas of the health system with multiple barriers to healthcare access. Beside distance from cities and towns, they also face administrative barriers arising from citizenship and movement to reach functional services—especially if from the other country. If not well articulated in policies and arrangements of health service provision, populations along the borders may remain vulnerable to poor access to healthcare and at risk of being left behind. This study set out to document experiences and feasible strategies to advance access and UHC goals especially for communities residing along the EAC borders. This study addressed two questions: (1) why border residents cross to seek healthcare across the border and (2) how they navigate institutional health system barriers and challenges of seeking healthcare across the border (physical, financial, availability and acceptability) in East Africa.

## METHODS
### Study setting and sites
This study was conducted in four land-cross-border sites in East Africa in the countries of Uganda, Rwanda and Kenya. These sites are: (1) Katuna-Uganda/Gatuna-Rwanda, (2) Rusomo-Rwanda/Lusomo-Tanzania, (3) Busia-Uganda/Busia-Kenya and (4) Isebania-Kenya/Sirare-Tanzania. The physical location of the study sights is shown in figure 1. These border sites are mostly remote and peasant with a few trading centres acting like border posts. Only Busia (Uganda-Kenya border) is a large urban trading site. At the formal border posts there are small business enterprises to promote cross-border trade. The sites are usually located within a district (Uganda and Rwanda) or a county (Kenya). The districts or county are served by a local health facility that should provide a minimum health package to the citizens as mandated by the government. At these local health facilities, the services are only planned and budgeted to serve the citizens. In Rwanda, all citizens have health insurance and one can only receive care under insurance. The study sites have some differences in the level of care that can be provided, for example, while in Busia-Uganda there is a health centre three functioning under a 'free healthcare policy', in Busia-Kenya, there is a hospital (is higher

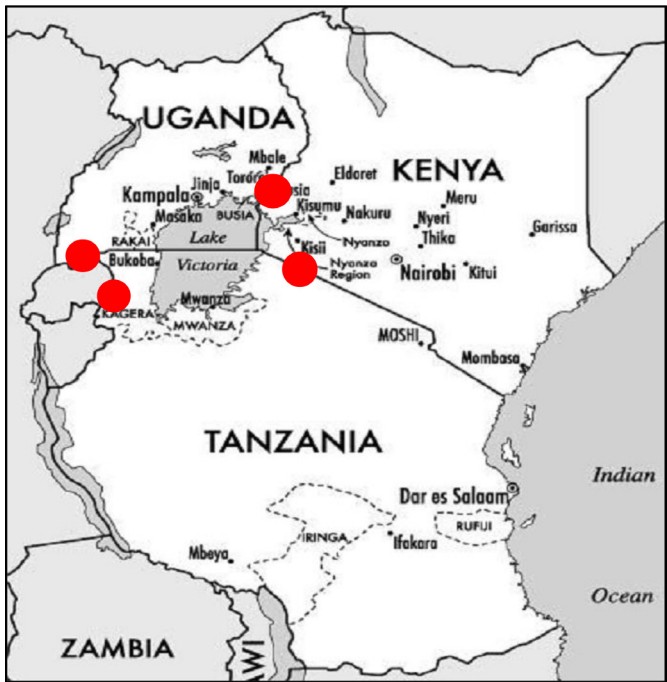

**Figure 1** Study sites (indicated by red circles).

than a health centre three) and some services can only be accessed through pay by non-citizens.

### Study design and respondent population

A cross-sectional design was used with quantitative and qualitative data collection methods. The study participants were the border resident communities at the selected border sites. We defined a border resident community as that community within 5–10 km on either side of a designated international border crossing point, depending on whether the community is clustered at the exact crossing point or along the main highway that transits through the border crossing point. This study collected data around there case conditions that were selected to provide diverse experience in terms of (1) chronic care (HIV treatment), emergency care (maternal delivery) and prevention (childhood immunisation). These three cases also represent typical services that are part of the healthcare packages in all East African countries.

### Sample size and sampling

At each study site, 75 survey respondents were targeted for survey and were divided equally across either side of the border for the three case conditions under study (ie, a total of 300) as a purposeful sample that could ensure we collected as many diverse views as possible given the resources that we had. This study realised a sample of 279 household adults. To identify potential survey respondents, we asked the in-charges of the major catchment health facility on each of the study site to invite household adults in the health facility catchment area for a community dialogue. During the dialogue, the household adults (or any of their household members) who had ever

accessed care from the opposite side of the border in the past 5 years were identified.

On identification, each of the household adult was secretly screened by the researcher to identify which care they or their household member(s) had accessed from the opposite side of the border. If the household adult or their household member(s) had ever accessed HIV treatment refill, maternal delivery and/or childhood immunisation services from the opposite side of the border within 5 years prior to this study, they were consented for an interview about their experience accessing care across the border. The process of confidential screening was repeated until the targeted number of respondents for each case condition was realised.

### Questionnaire development and data collection

From February to June 2018, an interviewer administered semistructured questionnaire was used to survey the respondents. During the interview, questions were sequenced[23] to first explore the narration of the respondents' experience seeking care followed by the Likert-style survey questions about the perceived ease of access across the health access dimensions and finally the reasons for the stated score.

A questionnaire was developed specifically for this study. The design was informed by the health access framework.[8] This framework as conceptualised by Levesque and Harris et al (2013) provides four interrelated domains for explaining the access pathway to healthcare by communities. These include geographical, affordability, accommodation and acceptability. The questionnaire was divided into four subsections corresponding to these four adapted dimensions of the health access. Each dimension contained between 5 to 16 Likert-scale (scale of 0–4) questionnaire items developed to cover the concepts central to that healthcare access dimension in the context of cross-border settings. This method is built around exploratory factor analysis (EFA) and it is recommended for assessing multi-dimensional concepts such as healthcare access in urban setting.[24 25] The questionnaire items for each subdimension were phrased to reflect the pathway of the client's decision to seek care across the border up to the time of receiving care or failing to receive care. Each dimension also had an overall Likert-scale item to assess overall contribution of that dimension to their recent experience of cross-border healthcare access.

The questionnaire had one open-ended question that asked each of the study respondents to give a narration of their most recent vivid experiences[26] seeking healthcare across the border. During the narration, we probed for factors that enabled navigation of barriers for example: cost of care, crossing requirements and quality of care. The narrations were audio recorded with consent of the respondents and transcribed verbatim.

The questionnaire content validation was explored by two rounds of workshops by the research team. The designed tool was pretested at one similar site in Uganda

before initiating data collection. Analysis of pretest data showed adequate (above 0.70) Cronbach's alpha reliability coefficients for the items in all dimensions of the tool after elimination of few questionnaire items. The tools were translated into the local language of the respective data collection site. The languages used included: Swahili (Kenya), Kinyarwanda (Rwanda), Rukiga (Uganda-Katuna border) and Lusyamya (Uganda-Busia border). As part of the translation process, the tool was translated from the original English version into the site-specific local language (Swahili, Kinyarwanda, Rukiga and Lusyamya). Back translation from local language versions back to English was used compared with the original version.[27] The interviews were collected by a team of experienced and well-trained young researchers on the study team.

### Data management and analysis
#### Quantitative data management and analysis
Filled questionnaires were checked for completeness, assigned a unique identifier and entered in *Epi Data*. The summary data set was then exported to *SPSS V.23* for analysis. In preliminary analysis, we determined the reliability of the items in each of the four subsections of the questionnaire separately using the Cronbach's alpha reliability coefficient

Next, we conducted dimension reduction using EFA in *SPSS V.23* to generate fewer and meaningful (latent) variables with eigenvalues of 1 and above. To retain variables that clearly discriminate between the generated latent variables, the minimum cut-off for item loading was set at 0.7.

Finally, a regression was performed to determine the independent association of each of the generated latent variables to the dependent variable—'perceived ease of access to healthcare' on the opposite side of the border. Beta coefficient, CI and p values were used to gauge the strength of association of latent factors with the dependent variable. Results were stratified by case condition and by dimension of the health access framework.

#### Qualitative data management and analysis
For the qualitative data, the recordings and all handwritten notes were kept in a secure location at the study secretariat office of the partnering universities that led the data collection. No identifying information of any of the specific respondents was included in the transcription and analysis reports. Qualitative data were exported to *Atlas TI version 8* for analysis. A conventional deductive content analysis approach was used.[28] From the text, codes were derived then sorted into categories then grouped together to inform emerging overarching themes. The analysis took into considerations any relevant comparisons in the different study sites, country where healthcare was sought, gender and case condition. Relevant qualitative quotes from respondents were presented in the results sections to support the key results under the respective themes and categories. In order to

ensure trustworthiness, each transcript was coded by two researchers and any differences were discussed by the research teams to reach consensus. The derivation of the subthemes and themes was done in a workshop with all the authors. The research team has a good mix of both qualitative and quantitative expertise.

### Patient and public involvement
Prior to the data collection field work, the study team conducted a study previsit meeting to all the study sites and engaged local leaders, health officials and border managers in order to introduce the study objectives and seek their permission and consent to participate in the study. The study team worked with some of the community members as field guides during the data collection. The study team held dissemination meetings at all the study sites in order to validate the findings.

## RESULTS
### Respondent characteristics
A total of 279 households were surveyed. The majority of respondents were women (n=251, 90%), and married (n=213, 76.3%). Only 4% (n=11) were in formal employment. The majority had relatives across the border in the neighbouring country 75% (n=207). The detailed characteristics of the respondents are summarised in table 1.

### Characteristics of individuals who accessed healthcare from the opposite side of the border
Majority of the people who had accessed healthcare from the opposite side of the border were women (n=205, 73.5%). Those accessing care for more than a year ago were 27.6% (77/279) while 22.9% (64/279) had accessed care in less than a month prior to interview. The majority accessed care from public health facilities (n=234, 83.9%), while 104 (37.3%) had sought immunisation services. The details are summarised in table 2.

### Reliability analysis of the survey questionnaire
The respondents were surveyed on four (4) dimensions of cross-border healthcare access. The reliability analysis of the survey questionnaire, reported as Cronbach's alpha showed the following results. Physical; 0.647, affordability; 0.215, availability and accommodation; 0.783, acceptability and appropriateness; 0.645.

### The most significant factors that facilitate successful access to healthcare from the opposite side of the border
Principal component analysis with varimax rotation was conducted in SPSS to generate latent variables for all the access dimensions. The eigenvalues greater than one rule and scree tests were used to decide the numbers of factors/latent variables to retain. The latent variables were labelled according to the items that loaded onto them the most.[29] For example, six new/latent variables were generated for the physical access dimension. The items that loaded onto the second latent variable in the physical access dimension are items C14 and C16. To decide

**Table 1** Sociodemographic characteristics of the household adults who reported a household member who had ever crossed the border to seek healthcare services

| Characteristic | Frequency (n) | Percentages (%) |
|---|---|---|
| Sex | | |
| Male | 28 | 10.0 |
| Female | 251 | 90.0 |
| Age (mean, SD) | Mean=32.8 | SD=9.8 |
| Educational level | | |
| No education | 35 | 12.5 |
| Some primary | 107 | 38.4 |
| Completed primary | 48 | 17.2 |
| Some secondary | 49 | 17.6 |
| Completed secondary | 27 | 9.7 |
| Tertiary | 13 | 4.7 |
| Marital status | | |
| Single | 28 | 10.0 |
| Married | 213 | 76.3 |
| Divorced | 16 | 9.3 |
| Widowed | 12 | 4.3 |
| Employment status | | |
| Not employed | 131 | 47.3 |
| Informal and self-employment | 135 | 48.7 |
| Formal employment | 11 | 4.0 |
| Nationality | | |
| Kenyan | 82 | 31.1 |
| Rwandese | 89 | 33.7 |
| Tanzanian | 6 | 2.3 |
| Ugandan | 86 | 32.6 |
| Dual citizenship | 1 | 0.4 |
| Has relatives on opposite border | | |
| Yes | 207 | 75.0 |
| No | 69 | 25.0 |
| Border site | | |
| Busia Kenya-Busia Uganda | 97 | 34.8 |
| Gatuna-Katuna | 91 | 29.0 |
| Isebania | 49 | 17.6 |
| Rusumu | 52 | 18.6 |
| Country of residence | | |
| Kenya | 84 | 30.1 |
| Rwanda | 101 | 36.2 |
| Uganda | 94 | 33.7 |

SD, Standard deviation.

**Table 2** Characteristics of border residents who accessed healthcare from the opposite side of the border

| Characteristic | Frequency (n) | Percentages (%) |
|---|---|---|
| Age (mean, SD) | Mean=22.7 | SD=16.7 |
| Sex | | |
| Male | 74 | 26.5 |
| Female | 205 | 73.5 |
| When care was sought | | |
| <1 month | 64 | 22.9 |
| 1–3 months | 58 | 20.7 |
| >3–6 months | 26 | 9.3 |
| >6–12 months | 54 | 19.3 |
| >12 months to 5 years | 77 | 27.6 |
| Type of facility where care was sought | | |
| Public | 234 | 83.9 |
| Private not for profit | 11 | 3.9 |
| Private for profit | 32 | 11.5 |
| Don't remember | 2 | 0.7 |
| Service received | | |
| Delivery | 98 | 35.1 |
| HIV treatment | 77 | 27.6 |
| Immunisation | 104 | 37.3 |
| Type of service received | | |
| Diagnosis only | 3 | 1.1 |
| Treatment at Outpatient Department | 192 | 68.8 |
| Admission | 40 | 14.3 |
| Operation | 44 | 15.8 |
| Reasons why care was sought | | |
| Better quality | 99 | 36.0 |
| Referral | 10 | 3.6 |
| Less costly | 79 | 28.7 |
| Community support | 18 | 6.6 |
| Stigma | 12 | 4.3 |
| Proximity | 25 | 9.0 |
| Other | 51 | 18.6 |

SD, Standard deviation.

the name of this new variable, we merged the meaning of C14 and C16 and called the variable 'Social Connections'. This was done for all the latent variables across the four (4) access dimensions. These findings are summarised in figure 2.

### Overall ease of cross-border access to healthcare

The Likert-scale median scores ranged from 3 to 4. Overall, all access dimensions were important factors in successful cross-border health access. For different case conditions however, different access dimensions had different levels of contribution to successful cross-border health access. For respondents who accessed maternal delivery services across the border, availability and accommodation dimension contributed more to their successful cross-border healthcare access. For respondents who accessed HIV treatment

**Physical access Dimension**

| Items | Latent variables/Item loadings | | | | | |
| | Ease of crossing the border | Social Connections | Satisfaction with organization of service at home | Referral Documents | Satisfaction with organization of services across the border | Distance to the health facility |
|---|---|---|---|---|---|---|
| c1 | | | | | | .860 |
| c2 | .702 | | | | | |
| c3 | .756 | | | | | |
| c5 | | | .873 | | | |
| c6 | | | .905 | | | |
| c7 | | | | | .857 | |
| c8 | | | | | .822 | |
| c10 | .728 | | | | | |
| c12 | | | | .751 | | |
| c13 | | | | .739 | | |
| c14 | | .770 | | | | |
| c16 | | .707 | | | | |

**Financial access/ Affordability Dimension**

| Items | Latent variables/Item loadings | | | | |
| | Self-funded | Time & transport cost acceptable | Cost of care affordable | Insurance needed | Local currency and financial support |
|---|---|---|---|---|---|
| d1 | | | .745 | | |
| d3 | | .875 | | | |
| d4 | | .832 | | | |
| d5 | .862 | | | | |
| d7 | -.824 | | | | |
| d9 | | | | | .793 |
| d10 | .825 | | | | |
| d11 | | | | .826 | |

**Availability and Accommodation Dimension**

| Items | Latent variables/Item loadings | | | | |
| | Better Workforce | Provider Public-private mix Across | Services & medicines available Across | Not getting better at home | Language was not a barrier |
|---|---|---|---|---|---|
| e1 | | | | 0.797 | |
| e2 | | | | 0.717 | |
| e4 | | | 0.842 | | |
| e5 | 0.855 | | | | |
| e6 | 0.825 | | | | |
| e8 | | | 0.791 | | |
| e10 | | | | | |
| e11 | | 0.709 | | | |
| e13 | | 0.718 | | | |
| e16 | | | | | 0.835 |

**Acceptability and Appropriateness Dimension**

| Items | Latent variables/Item loadings | |
| | Acceptable Quality | Accountability |
|---|---|---|
| f1 | .900 | |
| f2 | .911 | |
| f3 | | |
| f4 | | .858 |
| f5 | | .811 |

**Figure 2** New generated variables for each of the four studied access dimensions and corresponding items that loaded with a factor of 0.7 and above onto the new variables.

services, affordability contributed more. Service related findings are summarised in figure 3.

## Independent predictors of cross-border health access

Separate regressions were run against the mean score of the outcome variable 'overall ease of access of cross-border healthcare' for all the access dimensions. In all regressions, the independent variables were the latent variables extracted using factors analysis. The regression analysis was also stratified for each of the three case conditions—HIV treatment, childhood immunisation and maternal delivery.

For physical access, the model explained 36% variation, which varied between 38% and 40% depending on the stratification by the three case conditions. The strongest

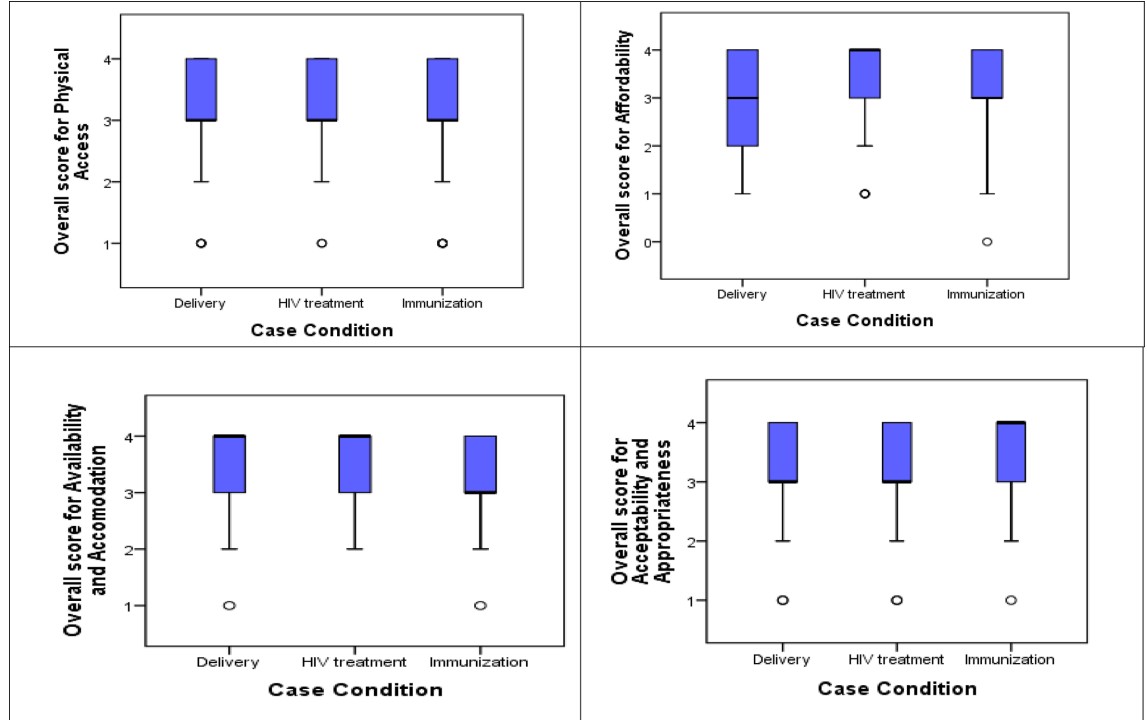

**Figure 3** The extent to which each access dimension contributed to the ease of accessing healthcare from the opposite side of the border for each of the three case conditions.

independent predictor of 'overall ease of physical crossing for care' was 'ease of border crossing'. For financial access/affordability, the model explained 31% variation, which varied between 21% and 44% depending on the case condition. Access to free care was (inverse of self-funding) was one of the main predictors. Table 3 provides the main results . Latent variables that were significant at a P-value less than 0.05 are bolded in the table 3.

## Qualitative findings

From the qualitative analysis, two overarching themes emerged and these were 1) reasons for crossing the border to seek healthcare and 2) experiences on how resident communities navigate barriers during cross-border healthcare access. The results are summarised in table 1 in the online supplemental file 1 attached.

## Reasons for crossing the border to seek healthcare

The main push factors for crossing to seek care across the border were: (1) service delivery points perceived as more accessible, (2) health services perceived as more affordable, (3) availability of services needed and (4) acceptability of the services.

### Physical accessibility of the health services

The border residents preferred to access services that are closer to them. In instances where the nearest health facilities was on the opposite side of the border, the residents preferred to cross the border to access services rather than travelling long distances to access services in their own country;

> The reason I went to Uganda, it is because the health facilities where we get our vaccination (RUBAYA, CYUMBA and KANIGA) are more distant, and the transport to reach there is expensive—Border resident, Gatuna-Rwanda.

Having relatives and friends on the opposite side of the border was mentioned as a reason why most border residents preferred to seek care across the border since they had an opportunity of being taken care of by their relatives:

> My relative live there, so I went there because they could take care of me after giving birth …. I just crossed because I knew my family would receive me—Border resident, Gatuna-Rwanda.

### Affordability of the health services

Health services being more affordable prompted study communities to seek healthcare by crossing the border. This was more common where national health insurance was established in the home country and someone wanted to access services that are not covered by insurance at home. They crossed to where there was a 'free services' policy. They further noted that accessing these services without insurance in their own country was more costly. One respondent noted:

> It all started when I was coming from the hospital … I needed medication to recover, but, the medicines I required, were not covered under my health insurance. So a friend of mine recommend me to go to Uganda side—Border resident, Gatuna-Rwanda.

### Availability of the health services

The inadequacies of services in one's home country often prompted border resident communities to cross. Border residents who visited the health facilities in their home countries and did not get the services they required tended to cross the border to access these services;

> I took my grandchild because after visiting the hospitals here for three consecutive times and missing the vaccine, I was forced to travel to Uganda for the vaccine—Border Resident, Busia-Kenya.

Formal and informal referral to quality, nearer and cheaper health facilities prompted border residents to seek healthcare from health facilities in the opposite side of the border. Formal referrals were arranged by a health worker while informal referrals were prompted by self, friends, relatives and other community members.

> I started feeling labor pains where I visited the district hospital but got students who were attending to patients. I was bleeding and was advised to go to MASAFU (Uganda side) for delivery—Border resident, Busia-Kenya.

### Acceptability of health services

This theme was related to the confidentiality of care sought across the border and perceived better quality of the care across the border. The fear of being stigmatised by fellow community members was also found among a few respondents. This prompted HIV positive border residents to seek healthcare from health facilities across the border fearing that home communities would come to know about their HIV serostatus

> When I discovered I was infected with HIV I was stigmatized and couldn't stand going to pick drugs as I could meet very many people I know. I was forced to go to Uganda and gained courage when picking the drugs—Border resident, Busia-Kenya.

Border residents who perceived that services from the opposite side of the border were of better quality than at home tried to seek healthcare from health facilities across the border;

> When I went to the Kenyan hospital they worked on me well and I came back safely. The good thing in Kenya is that any nurse can help you, she tells you where to start from, and she directs where to get the

**Table 3** Independent predictors of overall ease of cross-border access to healthcare

| Access dimension | Regression model items | Overall | | Delivery | | HIV | | Immunisation | |
|---|---|---|---|---|---|---|---|---|---|
| | | Regression coefficient | P value | Regression coefficient | P value | Regression coefficient | P value | Regression coefficient | P value |
| Physical access | Constant | 3.221 | 0.000 | 3.23 | 0.000 | 3.15 | 0.000 | 3.254 | 0.000 |
| | Ease of border crossing | 0.381 | 0.000 | 0.346 | 0.000 | 0.565 | 0.000 | 0.371 | 0.000 |
| | Social connections | −0.79 | 0.061 | −0.09 | 0.209 | −0.035 | 0.702 | −0.05 | 0.475 |
| | Satisfaction with organisation and approachability of facilities at home | −0.006 | 0.883 | −0.012 | 0.865 | 0.126 | 0.130 | −0.105 | 0.138 |
| | Documentation of referral for healthcare needs | 0.098 | 0.021 | 0.146 | 0.085 | 0.034 | 0.627 | 0.133 | 0.066 |
| | Satisfaction with the organisation and approachability of facilities across the border | 0.156 | 0.00 | 0.164 | 0.037 | 0.123 | 0.140 | 0.217 | 0.002 |
| | Distance to facility | 0.224 | 0.00 | 0.213 | 0.004 | 0.316 | 0.002 | 0.226 | 0.001 |
| | R square | 0.361 | | 0.379 | | 0.402 | | 0.388 | |
| | N | 251 | | 89 | | 69 | | 91 | |
| Financial access/affordability | Constant | 3.009 | 0.000 | 2.918 | 0.000 | 2.78 | 0.000 | 3.027 | 0.000 |
| | Self-funded | −0.478 | 0.000 | −0.496 | 0.000 | −0.646 | 0.000 | −0.387 | 0.000 |
| | Time and transport cost acceptable | 0.004 | 0.946 | −0.075 | 0.344 | 0.488 | 0.002 | −0.049 | 0.624 |
| | Cost of care affordable | 0.324 | 0.00 | 0.522 | 0.000 | 0.147 | 0.126 | 0.260 | 0.013 |
| | Insurance needed | 0.006 | 0.917 | 0.018 | 0.821 | −0.233 | 0.041 | 0.154 | 0.133 |
| | Local currency and financial support | −0.047 | 0.406 | −1.13 | 0.151 | 0.066 | 0.525 | −0.049 | 0.616 |
| | R square | 0.305 | | 0.437 | | 0.437 | | 0.213 | |
| | N | 248 | | 95 | | 62 | | 89 | |
| Availability/accommodation | Constant | 3.451 | 0.000 | 3.443 | 0.000 | 3.487 | 0.000 | 3.411 | 0.000 |
| | Better workforce | 0.106 | 0.002 | 0.045 | 0.461 | 0.216 | 0.001 | 0.108 | 0.075 |
| | Provider public–private mix | 0.064 | 0.063 | 0.047 | 0.442 | 0.002 | 0.978 | 0.103 | 0.114 |
| | Services/medicines available | 0.274 | 0.00 | 0.297 | 0.000 | 0.393 | 0.000 | 0.144 | 0.040 |
| | Not getting better at home | 0.04 | 0.242 | 0.023 | 0.723 | 0.063 | 0.224 | 0.033 | 0.580 |
| | Language was not a barrier | 0.021 | 0.549 | 0.011 | 0.850 | −0.01 | 0.864 | 0.061 | 0.349 |
| | R square | 0.242 | | 0.295 | | 0.471 | | 0.113 | |
| | N | 252 | | 90 | | 70 | | 90 | |

Continued

**Table 3** Continued

| Access dimension | Regression model items | Overall | | Delivery | | HIV | | Immunisation | |
|---|---|---|---|---|---|---|---|---|---|
| | | Regression coefficient | P value | Regression coefficient | P value | Regression coefficient | P value | Regression coefficient | P value |
| Acceptability and appropriateness | Constant | 3.395 | 0.000 | 3.37 | 0.000 | 3.311 | 0.000 | 3.462 | 0.000 |
| | Acceptable quality | 0.364 | 0.000 | 0.388 | 0.000 | 0.372 | 0.000 | 0.325 | 0.000 |
| | Accountability | 0.142 | 0.000 | 0.154 | 0.004 | 0.228 | 0.000 | 0.062 | 0.304 |
| | R square | 0.364 | | 0.462 | | 0.396 | | 0.246 | |
| | N | 262 | | 92 | | 73 | | 95 | |

book and notes all things concerning the baby.—Border resident, Busia-Uganda

### Experiences on how border resident communities navigated barriers during cross-border healthcare access

From the narration of experiences with household adults that had successfully sought care across the border, the key facilitators that enabled navigation of barriers emerged across the physical and affordability dimensions. Under physical dimension, the main facilitators were presence of informal routes, having official travel documents and ability to speak a similar language. While under affordability the main facilitator was the ability to meet the cost of care across the border.

#### Presence of informal routes

The presence of informal routes locally known as '*panya roads*' facilitated access to healthcare services from the opposite side of the border. Many border resident communities were not allowed to cross at the formal crossing routes for health services and resorted to using informal routes. We also found a common ethnic identify of communities separated by the national borders. These communities speak the same local language across the border, they use both formal and informal routes to cross the border for health, markets and other reasons. Many community members have informal connections or relatives across borders but also are capable of independent and informal navigation of the healthcare access that bypass formal border controls and protocols.

> When I pass via the customs offices, they sometimes ask for money or arrest me but I was arrested once. Now, I normally pass via a panya (short cut route) (to Kenya side) to pick my medicine—Border resident, Busia-Uganda.

> Border officers send you back because all those services are available here….(in Rwanda). Actually, most people who need those services use shortcuts to cross. Furthermore, Ugandans who take HIV treatment here (in Rwanda) also pass through there (the informal route)—Border resident, Gatuna- Rwanda

#### Having official travel documents

Border residents who had their identification documents, such as; identity cards and passports were easily allowed to cross the border. Most of the border residents without travel documents were not allowed to cross the border and, in most instance, they had to make informal payments or bribes so as to be allowed to cross. Some respondents however mentioned that in some instances, travel documents were not a requirement for crossing the border and this facilitated healthcare access. One respondent remarked:

> No, whenever I visit there I am never asked for any document either when going or coming back—Border Resident, Busia-Kenya.

## Similarity in language

Knowing the local dialect of the neighbouring border communities facilitated access to healthcare services across the border. Border residents were able to easily communicate with the service providers and even to disguise as nationals of the destination country. On the other hand, the inability to speak the local language hindered access to healthcare services due to language barrier and sometimes had to get translators.

> I was just treated well…It's only the language barrier that made the whole process look tiresome—Border Resident, Isebania-Kenya.

## Affordability of cost of care

The cost of care included the money paid for the actual health services, the transport costs and other informal payments or bribes made to facilitate the process of accessing care. The respondents who had crossed to access care on the opposite side of the border unanimously agreed that their ability to pay the actual cost of services enables them to access health services from there. In most cases, HIV treatment and childhood immunisation services were free and these enabled border residents with these conditions to access care.

> The cost was so much reasonable as compared to Kenyans side and we also used Kenyan currency which is stronger that the Uganda shillings—Border Resident, Busia-Kenya.

Relatedly, being able to pay informal payments/ bribes either at the border point (to border officials) or at the health facility enabled access to health services at the opposite border

> While on the way we encountered the police whom we bribed to proceed with our journey to Dabani Hospital. At the hospital we had a midwife who was attending to her by bringing her food and water to bath and we ended up paying her (the midwife)—Border Resident, Busia-Kenya.

## DISCUSSION

This study aimed to explore how to increase healthcare access for communities along the border in East Africa. We found that communities that are resident near national borders were able to cross borders to seek healthcare. There are factors that attract border residents to seek care across the border and these centred on service delivery points being near, availability of services and health services being of acceptable quality across the border compared with the home country. From the quantitative analysis the main facilitators for cross-border health access were the ease of physically crossing and health services being affordable across the border among other factors. In general, the qualitative and quantitative results were in agreement and these are discussed together below.

Rather than travelling long distances to access healthcare in the home country, border residents preferred to access services that were closer to them though across the border. This has also been seen elsewhere[30] that patients gravitate to their nearest convenient health service provider. Other than physical 'closeness' of the healthcare across the border, many border residents in our study commonly used informal routes *Panya roads* to cross for healthcare—an indication of porous borders referred to as "ease of border crossing" in the liner regression analysis . The ease of physical access is also enhanced by the fact that for many of the border communities, the culture and language is the same and they live as one community irrespective of the border.[17] This is evidenced from the respondents' remarks that knowledge of the local language and family relatives that reside across the border facilitated crossing for healthcare.

Border residents chose to seek healthcare across the border because it was still affordable compared with care within their home country. This is in concordance with the linear regression analysis where the most important factor in the affordability dimension was the inverse of 'self-funded' since most services are usually free especially for the case conditions explored in our study. Studies done among US residents living along the USA–Mexico border have shown that due to the economic, financial and healthcare access barriers, border residents on the US side commonly resort to crossing into Mexico to meet their healthcare needs because of much more affordable prescription medications as well as services.[6 7] For the border residents in East Africa, there seemed to be an opportunity cost they were willing to pay in terms of other costs like transport and informal payments at border crossing. However, we acknowledge that this opportunity cost was easier for the border residents to bear since they were seeking services like childhood immunisation and HIV treatment refill which are usually free of charge in East Africa.[31] Apart from affordability in terms of cost of care, border residents who lacked health insurance also preferred to seek care across the border where national health insurance was not being implemented. There is need for East African countries to have common arrangements for example health insurance with portable benefits (or free care) that especially help poor communities along the border to access care. Although currently the non-insured community members can access care across the border, this leaves them disadvantaged in case the service needed is only available in their home country.[32]

Some border residents also preferred to go where care was perceived to be acceptable in terms of quality and confidentiality. This was more important for mothers seeking delivery services and patients with HIV that preferred confidentiality. This speaks to the need for countries to improves service quality so that citizens in remote areas are more willing to use the available services at home.[33]

From our study, we see the need to move from nationally circumscribed rights and entitlements to a legal environment that allows border residents to freely access care cross the border within a well-established and agreed on arrangement between countries. Although border residents are individually navigating access barriers, this leaves them vulnerable if some legal frameworks were to be enforced. For UHC to be efficiently delivered, countries may need to agree on and establish planning, resourcing and regulatory arrangements to enable border communities access the existing healthcare provision without bearing the hardship of navigating these barriers on their own.[22] This can also reduce the uncertainty and discretion of front-line officials if they were all aware of the access rights for border residents in neighbouring communities. This can span to other cross-border health issues such as having joint efforts to deal with disease outbreaks and pandemic control such as COVID-19.[34 35]

At regional level, centres of excellence for healthcare are beginning to form with a view of serving people from the EAC.[36] These will require innovative cross border policies and arrangements for efficient delivery of healthcare. It would be useful to share the available healthcare resources (eg, every side of the border does not have to build a hospital if one side of the border has one in close distance). Guidelines should be developed to enable the use of a common pool of healthcare resources and optimise efficiency on both sides of the border. The European region has tried to remove legal and administrative barriers through cross-border cooperation (CBC)[37] aimed at 'identifying potential complementarities in all fields of human activities in the border regions to ease the lives of the border residents'. There are lessons that can be drawn from these and other cross-border integration initiatives.[13] During the design of any CBC initiatives, host communities should be engaged in the plans as part of the efforts to promote 'tolerance, understanding and good neighbourly relations between populations'.[38]

As this paper goes to publication, cross-border health has been made more urgent by the COVID-19 pandemic and its impact on the border health systems - to provide testing and surveillance for interstate truck drivers in East Africa. The guidelines to manage the pandemic have come into sharp confrontation with the objectives for legal rights of interstate travel and movement of goods—in some cases threatening to dismantle the regional cooperation mission.[39] Numerous media carried stories showing long queues of trucks some extending 20 km at the border all waiting for coronavirus testing that was sometimes taking 2–3 days to offer the test results and clear the truck through emigration points. Communities around these long jams of trucks were also exposed to community transmission of coronavirus from the truck drivers—a group that had a high occupational risk of contracting and transmitting the virus.[40] This further demonstrates the need to invest in and coordinate better the economic and healthcare objectives at border points for the EAC to consolidate its integration and social economic functionality.

## Study limitations

This paper only presents the views of respondents that successfully navigated access barriers. Our study sites were the official cross points. We did not include the numerous informal cross points which may have different contexts and experiences. The use of official crossing was selected to enable the study to factor in the operations of the formal rules for emigration when seeking healthcare across the border.

## CONCLUSION

From this study, we note that communities resident near national borders of the EAC were able to cross borders to seek healthcare. Although the EAC integration agenda has been discussed for a while, progress has been made in the areas of economics and trade and it has not yet benefited border residents in terms of access to healthcare across the border. There is need for: innovative arrangements to optimise the existing resources and a policy environment to enable East Africa invest better and realise synergies for these communities for health integration in the EAC as well as for universal health coverage aspirations.

**Author affiliations**
[1]Department of Health Policy Planning and Management, School of Public Health, Makerere University, Kampala, Uganda
[2]Makerere University School of Public Health, SPEED Project, Kampala, Uganda
[3]Department of Health Policy Planning and Management, Makerere University School of Public Health, Kampala, Uganda
[4]Health Economics and Policy, African Health Economics and Policy Association, Kampala, Uganda
[5]College of Medicine and Health Sciences, University of Rwanda, Kigali, Rwanda
[6]Department of Health Policy and Management, College of Health Sciences, Moi University, Eldoret, Uasin Gishu, Kenya

**Acknowledgements** The authors thank the District or County Leadership at the following border sites: Katuna-Uganda (Kabale District); Gatuna-Rwanda (Gicumbi District); Rusomo-Rwanda (Kibehe District); Busia-Uganda (Busia District); Busia-Kenya (Busia County); Isebania-Kenya (Migori County). Thanks to Makerere University School of Public Health, SPEED Project, and many thanks to the following persons for their key role in the data collection activities: Mr Moses Sitati, Researcher at Moi University who led the field team in Kenya, Shadat Kititi of Makerere University School of Public Health who led the field team in Uganda and Vincent Mutabazi of University of Rwanda who led the field team in Rwanda.

**Contributors** FS, DT, SS, SB, AS, LM and MN conceptualised the study. FS, DT and SB led the team's field work. FS, LM and MN sought country level study approvals. All authors led the study previsits and data collection. FS, SS, MA and AS led the quantitative data analysis while FS, DT, SB, AR, LM and MN led the qualitative data analysis. FS, DT, SS and MA wrote the draft manuscript, with all authors providing feedback and approving the final version. All the authors take responsibility for their contributions.

**Funding** This work was supported by MRC/Wellcome Trust, UK, grant number: MR/R02028/1.

**Map disclaimer** The inclusion of any map (including the depiction of any boundaries therein), or of any geographic or locational reference, does not imply the expression of any opinion whatsoever on the part of BMJ concerning the legal status of any country, territory, jurisdiction or area or of its authorities. Any such expression remains solely that of the relevant source and is not endorsed by BMJ. Maps are provided without any warranty of any kind, either express or implied.

**Competing interests** None declared.

**Patient consent for publication** Not applicable.

**Ethics approval** Ethical approval for this study was provided by Makerere University Higher Degree Research and Ethics Committee (HDREC Protocol number 583) and the Uganda National Council of Science and Technology (Reg No SS4713). In Kenya the study was approved by Moi University/Moi Teaching Referral Hospital Research and Ethics Committee (FAN: IREC: 3077); in Rwanda was approved by the Rwanda National Ethics Committee (No. 438/RNEC/2018) and in Tanzania by the University of Dar es Salaam Research Ethics Committee (Reference Number UDSM-REC/2018/02). All study respondents provided informed consent prior to participating in the study.

**Provenance and peer review** Not commissioned; externally peer reviewed.

**Data availability statement** Data are available upon reasonable request. The datasets used and/or analyzed for our study are available from the corresponding author upon reasonable request.

**ORCID iDs**
Doreen Tuhebwe http://orcid.org/0000-0002-9464-2340
Aloysius Ssennyonjo http://orcid.org/0000-0002-5790-4874

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
