## [Reviewer comments · BMJ Open]

ARTICLE DETAILS

TITLE (PROVISIONAL)	Experiences of seeking health care across the border: Lessons to inform upstream policies and system developments on cross border health in East Africa
AUTHORS	Ssengooba, Freddie; Tuhebwe, Doreen; Ssendagire, Steven; Babirye, Susan; Akulume, Martha; Ssenyonjo, Aloysius; Rutaroh, Arthur; Mutesa, Leon; Nanagami, Mabel

VERSION 1 – REVIEW

REVIEWER	Seddighi, Hamed University of Social Welfare and Rehabilitation Science
REVIEW RETURNED	18-Dec-2020

GENERAL COMMENTS	This manuscript reviews experiences of seeking health care across the border in East Africa. In my opinion, this manuscript needs some minor revision. I include my specific recommendations below. 1. The way the method is written is confusing. I suggest separating the quantitative and qualitative methodologies and mentioning the statistical population, sampling, analysis method, validity and reliability, and other cases separately for each. In general, in my opinion, the qualitative method as well as its findings are poorly written.2. Regarding the context, very little explanation is given in the article and the reader outside the study area, can hardly understand the situation of the region in terms of socio-economic and medical access. I suggest authors write more about the context in terms of population, economy, and access to health care.3. The authors did not elaborate on how the questionnaire was constructed and how its variables were identified. I suggest that more information be provided about the construction of the questionnaire and that a questionnaire be attached for the reviewers.4. Using a qualitative research method before answering a research question through a quantitative method can help to create a questionnaire. Doing it after a quantitative study can also help to better discuss and interpret the findings of quantitative research. Please, in the manuscript, the authors specify the necessity of doing these methods simultaneously to answer the research question.5. Using tables and figures to present findings, especially in the qualitative findings section, can help readers better understand the content. For example, quotes could be mentioned in the table
--

	and themes and sub-themes could be presented in a figure in the article. 6. This research is based on the views of people who have crossed the border for medical services. However, the views of the host communities are not presented. I suggest the authors discuss in the discussion section the challenges and opportunities that host communities may face as a result. These challenges and opportunities may be mentioned in other articles, and some of them are to be expected. For example, host communities have not a problem for people coming across the border because of their medical care? Is there no protest in this regard? Given that hospital beds, treatment staff, etc. are provided for a specific town, is there no shortage of facilities with the presence of new people?
--	--

REVIEWER	Durham, Jo Queensland University of Technology, School of Public Health & Social Work
REVIEW RETURNED	02-Apr-2021

GENERAL COMMENTS	Thank you for the opportunity to review this paper Overall, the abstract needs to be rewritten it's very hard to follow you could more specifically highlight early on you are looking at residents who cross borders for healthcare and can be more specific about the countries otherwise your sample size looks small. Also mention in the abstract you used quantitative and qualitative methods The paper should also be edited to be more succinct Much of the cited literature about healthcare across borders is from Europe, there is also a broader range of relevant literature related to South-south travel which is relevant for this paper Introduction Overall, the introduction is clear but could be more succinct but It seems in the introduction that some of the barriers to care are the same in and outside of the country of residence In general, however you make the case for why people may cross borders and why we need to care but you don't mention state obligations – doesn't the state have an obligation to ensure quality and accessible healthcare to all its citizens? What is missing and might be important is how HCs are financed in these countries and can residents on the borders cross into the bordering country and receive health services at the same costs as residents of that country e.g. If I am from Rwanda can I cross into Uganda and pay the same as a Ugandan? Methods Can you be more specific as to why you selected the 3 conditions – e.g. are they specific to the region? Are they ones where access is particularly poor/high unmet need? Etc Were the items on the questionnaire developed specifically for this study? Can you explain how the instrument was "The questionnaire was validated through several rounds of iterations between the principal investigators and the 3 co-investigators of the study."
---

What was the process of translating into the local language? What about translating Lusoga? And is the official language of Uganda? And is it a written language

Why 75 participants per site?

Did you have ethnics approval to identify potential participants who had ever accessed care from the opposite side of the border in the past five years in a group dialogue (meeting)? Was there any change of someone whose HIV status was unknown to the community to become open in this process?

Please explain how potential participants were randomly sampled

How many people did the qual analysis? What measures were put in place to ensure trustworthiness?

Findings

Under findings you say – “This prompted HIV positive border residents to seek health care from health facilities across the border fearing that home communities would come to know about their

HIV sero-status” yet you categorised participants in community dialogue

How did participants ‘disguise as nationals of the neighbouring country’?

Some of the themes are very “thin” e.g Presence of informal routes, can you provide more detail, based on how you described your methods you should have rich data

Were there brokers or did people cross the border independently?

Were the border areas rural or urban

Limitations

Are there other limitations of your research related sampling or perspective of HC providers?

What about health outcomes?

Any limitations of the quantitative methods?

Cronbach’s alpha for Affordability; 0.215 – were you satisfied with this?

Discussion

Overall, this section could be more analytical and less descriptive

The interpretations from the quantitative component do not seem to be included or integrated into the discussion

What would common arrangements (insurance or free care) look like? How would they be financed? What about costing and pricing and cost-recovery, quality of care/standards – would governments be able to “sell” this to their citizens

‘Some border residents also preferred to go where care was of acceptable quality’ what did they not like about the quality of services in their place of residence? The statement about HIV seems to relate more to fear about being stigmatised or shame

‘From our study, we see the need to move from nationally circumscribed rights and entitlements to a legal environment that allows border residents to freely access care cross the border if the nearest service delivery point is on the opposite side of the border’ – so do you mean people are illegally crossing the border for HC?

Surely this is not new – what are government positions on this?

	‘For UHC to be efficiently delivered, countries may need to revise their planning, resourcing and regulatory arrangements to enable border communities access the existing healthcare provision without bearing the hardship of navigating these barriers on their own. This can also reduce the uncertainty and discretion of front-line officials if they were all aware of the access rights for border residents in neighboring communities” – not sure you have presented evidence to convince the reader this is an issue
--	---

REVIEWER	Aiura, Hiroshi Nanzan University, Faculty of Economics
REVIEW RETURNED	20-Apr-2021

GENERAL COMMENTS	The report on "Experiences of seeking health care across the border: Lessons to inform upstream policies and system developments on cross border health in East Africa" (bmjopen-2020-045575). Summary This paper explores the factors to access to health care services across the border, by a cross-sectional survey among 279 household adults. The linear regression analysis shows that the main predictors for ease of access for health care were "ease of border crossing", "small self-foundation", "Services and medicines availability" and "acceptable quality of services". The deductive content analysis shows the key facilitators for successful navigation of access barriers such as the presence of informal routes, speaking a similar language. Comments  1. The results the abstract described have not been in the body of the paper. Especially, the abstract shows that nearly one third (77/279) of the respondents had accessed care from the opposite side of the border, but I could not find this result in the body of the paper. Therefore, I did not know whether the results of the paper are reliable. 2. I could not find (policy) implications based on the results of the linear regression analysis. I feel that the results of the linear regression analysis cannot sufficiently resolve the issues of this study. 3. If the aim of the study is to find the reasons why border residents cross or fail to cross the border, I am afraid that the author(s) should compare the group of the residents crossing the border with the group the residents not crossing. 4. I could not judge whether the labels of latent variables are appropriate in the factor analysis, because I do not know what the list of questionnaires is. 5. The paper did not show P values for each latent variable. If P values were high, latent variables would not statistically influence ease of access for health care.
---

6. "figure 1" in "Overall ease of cross border access to healthcare" at page 8 would be a typo of "figure 3".

VERSION 1 – AUTHOR RESPONSE

Reviewer: 1

Dr. Hamed Seddighi, University of Social Welfare and Rehabilitation Science

Comments to the Author:

This manuscript reviews experiences of seeking health care across the border in East Africa. In my opinion, this manuscript needs some minor revision. I include my specific recommendations below.

1. The way the method is written is confusing. I suggest separating the quantitative and qualitative methodologies and mentioning the statistical population, sampling, analysis method, validity and reliability, and other cases separately for each. In general, in my opinion, the qualitative method as well as its findings are poorly written.

Response: We have added some subheadings (line 152, 169, 209, 226) in the methods to improve on the clarity of the section while keeping the qualitative and quantitative writups together. This is because the quantitative and qualitative methods are thematically similar hence written in a complementary way. Access issues were explored both quantitatively and qualitatively with open ended questions. Separating them will create repetitions for the readers and increase the length of the paper. In our view the quantitative and qualitative methods are better presented together for coherence and complementarity of the findings.

2. Regarding the context, very little explanation is given in the article and the reader outside the study area, can hardly understand the situation of the region in terms of socio-economic and medical access. I suggest authors write more about the context in terms of population, economy, and access to health care.

Response: The section on "study setting and sites" has been revised to explain more about the context as shown on page 5, Line 126-140. Given the many sites under study the space limited the extent to which the context can be described. However, we have highlighted the key factors relevant to the study

3. The authors did not elaborate on how the questionnaire was constructed and how its variables were identified. I suggest that more information be provided about the construction of the questionnaire and that a questionnaire be attached for the reviewers.

Response: The questionnaire was developed on the basis of the healthcare access framework. The framework helped to identify the access constructs and the variables. These were operationalized for cross-border settings. This has been clarified on page 6, line 175-189. We have also attached a copy of the operationalized questionnaire.

4. Using a qualitative research method before answering a research question through a quantitative method can help to create a questionnaire. Doing it after a quantitative study can also help to better discuss and interpret the findings of quantitative research. Please, in the manuscript, the authors specify the necessity of doing these methods simultaneously to answer the research question.

Response: The study communities are very hard to reach logistically since they are spread over a long distance hence expensive to visit the sites multiple times. Our study applied sequence in the process of interviewing as indicated on page 6, line 170-174. Respondents were invited to score the access issues and then asked to give reasons for their scores. In addition, many studies explore the health access concepts using qualitative and quantitative methods simultaneously if a researcher can access a population. This approach is well established in the literature. We have added the references 24 and 25 to this effect for the section on questionnaire development in the manuscript.

5. Using tables and figures to present findings, especially in the qualitative findings section, can help readers better understand the content. For example, quotes could be mentioned in the table and themes and sub-themes could be presented in a figure in the article.

Response: We have added a table (table 1 below and as supplementary file 1) to show the emerging themes and illustrative quotes as proposed by the reviewer.

Reasons for crossing the border to seek health care		
Theme	Sub themes	Illustrative quote
****Physical accessibility of the health services	Nearest health facilities is on the opposite side of the border	“The reason I went to Uganda, it is because the health facilities where we get our vaccination (RUBAYA, CYUMBA and KANIGA) are more distant, and the transport to reach there is expensive” -- Border resident, Gatuna-Rwanda.
	Having relatives and friends on the opposite side of the border	“My relative live there, so I went there because they could take care of me after giving birth I just crossed because I knew my family would receive me”-Border resident, Gatuna-Rwanda.
***Affordability of the health services	Health services more affordable across the border	“It all started when I was coming from the hospital ... I needed medication to recover, but, the medicines I required, were not covered under my health insurance. So a friend of mine recommend me to go to Uganda side”- Border resident, Gatuna-Rwanda.
**Availability of the health services	Presence of needed services across the border	“I took my grandchild because after visiting the hospitals here for three consecutive times and missing the vaccine, I was forced to travel to Uganda for the vaccine”- Border Resident, Busia-Kenya.

	Formal referral by health workers at home	"I started feeling labor pains where I visited the district hospital but got students who were attending to patients. I was bleeding and was advised to go to MASAFU (Uganda side) for delivery"- Border resident, Busia-Kenya.
	Informal referrals by self, friends and other community members	"...when I was sick, my employer she asked if I could have caught HIV. Therefore, she sent me to MAYENGO (Uganda side) for diagnosis, and I found that I had HIV"--Border resident, Busia-Kenya.
*Acceptability of health services	Confidentiality of care sought across the border	"When I discovered I was infected with HIV I was stigmatized and couldn't stand going to pick drugs as I could meet very many people I know. I was forced to go to Uganda and gained courage when picking the drugs"-Border resident, Busia-Kenya.
	Perceived better quality of care across the border	"When I went to the Kenyan hospital they worked on me well and I came back safely. The good thing in Kenya is that any nurse can help you, she tells you where to start from, and she directs where to get the book and notes all things concerning the baby."- Border resident, Busia-Uganda
Experiences on how border resident communities navigated barriers during cross border health care access		
Theme	Sub themes	Illustrative quote

****Ability to physically cross into the neighbouring country	Presence of informal routes	“Border officers send you back because all those services are available here....(in Rwanda). Actually, most people who need those services use shortcuts to cross. Furthermore, Ugandans who take HIV treatment here (in Rwanda) also pass through there (the informal route)” – Border resident, Gatuna- Rwanda
	Official travel documents	“No, whenever I visit there I am never asked for any document either when going or coming back”- Border Resident, Busia-Kenya.
	Similarity in language	“I was just treated well...It’s only the language barrier that made the whole process look tiresome”- Border Resident, Isebania-Kenya.
****Ability to afford care across the border	Ability to meet direct cost of care	“The cost was so much reasonable as compared to Kenyans side and we also used Kenyan currency which is stronger than the Uganda shillings”-- Border Resident, Busia-Kenya.
	The indirect costs of care	“While on the way we encountered the police whom we bribed to proceed with our journey to Dabani Hospital. At the hospital we had a midwife who was attending to her by bringing her food and water to bath and we ended up paying her (the midwife)”-Border Resident, Busia-Kenya.

6. This research is based on the views of people who have crossed the border for medical services. However, the views of the host communities are not presented. I suggest the authors discuss in the discussion section the challenges and opportunities that host communities may face as a result. These challenges and opportunities may be mentioned in other articles, and some of them are to be expected.

For example, host communities have no a problem for people coming across the border because of their medical care? Is there no protest in this regard? Given that hospital beds, treatment staff, etc. are provided for a specific town, is there no shortage of facilities with the presence of new people?

Response: We have discussed the implication of the findings to the host community as indicated in the discussion section on page 16, line 507-510. We are not able to effectively represent these views in this paper due to space limitations. The concerns of the host will be published in a separate paper drawing on the discussion at the validation meetings that were held to share the findings at the study sites.

Reviewer: 2

Dr. Jo Durham, Queensland University of Technology

Comments to the Author:

Thank you for the opportunity to review this paper

1. Overall, the abstract needs to be rewritten it's very hard to follow you could more specifically highlight early on you are looking at residents who cross borders for healthcare and can be more specific about the countries otherwise your sample size looks small. Also mention in the abstract you used quantitative and qualitative methods

Response: The abstract has been revised to indicate the countries where the study was conducted and to reflect that we used both qualitative and quantitative methods. Page 1 line 24-27

2. The paper should also be edited to be more succinct

Response: The article has benefited from editing service of an English language expert.

3. Much of the cited literature about healthcare across borders is from Europe, there is also a broader range of relevant literature related to South-south travel which is relevant for this paper

Response: We acknowledge this limitation. The literature on south-south travel is related to seeking high end treatment for specialized care, sometimes on control of epidemic prone diseases and less on seeking routine health services like the ones explored in our study. We hope our study builds on this body of knowledge.

Introduction

5. Overall, the introduction is clear but could be more succinct but It seems in the introduction that some of the barriers to care are the same in and outside of the country of residence

Response: The introduction has been edited to be more succinct and to highlight why remote communities often times find the need to cross the border to seek healthcare

6. In general, however you make the case for why people may cross borders and why we need to care but you don't mention state obligations – doesn't the state have an obligation to ensure quality and accessible healthcare to all its citizens?

Response: The obligations of state as detailed in another paper that is being published separately. We have added a few sentences to state the importance of state obligation as shown in the introduction on page 3, line 78-80

7. What is missing and might be important is how HCs are financed in these countries and can residents on the borders cross into the bordering country and receive health services at the same costs as residents of that country e.g. If I am from Rwanda can I cross into Uganda and pay the same as a Ugandan?

Response: We have included this contextual information in the methods under study setting page 5, line 126-140

Methods

8.Can you be more specific as to why you selected the 3 conditions – e.g. are they specific to the region? Are they ones where access is particularly poor/high unmet need? Etc

Response: These are typical services that are part of the national care package and the three case conditions were selected to provide a diverse experience in terms of 1) prevention (childhood immunization), 2) chronic care (HIV treatment) and 3) emergency care (maternal delivery). This has been highlighted under the methods section on page 5, line 147-151

9.Were the items on the questionnaire developed specifically for this study?

Can you explain how the instrument was “The questionnaire was validated through several rounds of iterations between the principal investigators and the 3 co-investigators of the study.”

Response: This sentence has been revised to include the workshops to address content validity, pretest and the calculated reliability coefficient of the questionnaire items using Cronbach’s alpha as per the methods and findings. Please see page 7, line 197-201

10.What was the process of translating into the local language? What about translating Lusoga? And is the official language of Uganda? And is it a written language

Response: We have detailed out the process used to translate the tool into local languages spoken at the study sites as shown on page 7, line 201-207

11.Why 75 participants per site?

Response: The 75 participants per site was a purposeful sample targeted to ensure we collected as many diverse views as possible given the resources that we had. For each of three specified healthcare conditions (for HIV treatment, child immunization and maternal delivery), we purposed to identify at least twenty respondents at each study site. We did not use the representative sample approach. We have added the rational in the methods section on page 6, line 153-156

12.Did you have ethics approval to identify potential participants who had ever accessed care from the opposite side of the border in the past five years in a group dialogue (meeting)? Was there any change of someone whose HIV status was unknown to the community to become open in this process?

Response: We received ethical approval for the study in all countries (Uganda, Kenya, Tanzania and Rwanda). During the community dialogue, no discussion was focused on personal level illness such HIV. The participant selection process has been expounded on page 6, line 156-168

13.Please explain how potential participants were randomly sampled

Response: Community members were randomly invited to a community dialogue and we screened for those who had ever crossed for care. From the ones that had ever crossed, (potential respondents), the researchers privately screened each community member by asking each one of them secretly (self-report) in order to identify those that received care for HIV treatment, child immunization and maternal delivery i.e. criterion sampling. Upon identification of the case, the researcher noted the targeted persons and consented them for an interview. This has been explained on page 6, line 156-168

14.How many people did the qual analysis? What measures were put in place to ensure trustworthiness?

Response In order to ensure trustworthiness, each transcript was coded by two researchers and any differences were discussed by the researcher team to reach consensus. The derivation of the sub-themes

and themes was done in a workshop of all the research teams. The research team has a good mix of both qualitative and quantitative expertise. This process had been highlighted in the methods under data analysis on page 8, line 236-239

Findings

15. Under findings you say – “This prompted HIV positive border residents to seek health care from health facilities across the border fearing that home communities would come to know about their HIV sero-status” yet you categorised participants in community dialogue

Response: Community members were randomly invited to a community dialogue and we screened for those who had ever crossed for care. From the ones that had ever crossed, (potential respondents), the researchers privately screened each community member by asking each one of them secretly (self-report) in order to identify those that received care for HIV treatment, child immunization and maternal delivery i.e. criterion sampling. Upon identification of the case, the researcher noted the targeted persons and consented them for an interview. This has been explained on page 6, line 156-168. We did not openly screen for the reasons why the residents crossed to seek care across the border hence confidentiality was not breached.

16. How did participants ‘disguise as nationals of the neighbouring country’?

Response: The phrase has been revised to indicate that the residents disguised as nationals of the “destination” country on Page 14, line 413. Most cross-border communities have social kinships across the border and speak the language. In many cases the border is made up of natural landmark like a river with many formal and informal crossing points.

17. Some of the themes are very “thin” e.g. Presence of informal routes, can you provide more detail, based on how you described your methods you should have rich data

Were there brokers or did people cross the border independently?

Response: Since most people would speak the local language across the border, they just walked through straight without necessarily needing any assistance. We recognize presence of informal connections or relatives across borders who can act like brokers that can help one to cross but, these are few cases. Most of the resident communities across borders have common identities, speak the same language and it is always easy to communicate across borders. This context has been added in the findings on page 13, line 385-393

18. Were the border areas rural or urban

Response: They are basically rural with small trading centers apart from Busia (Uganda-Kenya border). This context has been highlighted in the section on study setting on page 5

Limitations

19. Are there other limitations of your research related sampling or perspective of HC providers?

What about health outcomes?

Any limitations of the quantitative methods?

Response: The appreciative inquiry approach that we used in this study certainly has limitations given that we engaged border residents with the experience which gave more of the story for those who made it. Because of the purposefulness nature of the sample, we are not able to represent experiences of the complex border environments across the region. These two points are represented in the section on study limitation on page 17, line 527-531

20. Cronbach's alpha for Affordability; 0.215 – were you satisfied with this?

Response: We recognize that the Cronbach's alpha for Affordability; 0.215 was low and we were satisfied with it because there were no affordability problems experienced by the respondents since most of the services explored were free-hence, no variance in affordability. This has been highlighted in the discussion section on page 15, line 460-463

Discussion

21. Overall, this section could be more analytical and less descriptive

The interpretations from the quantitative component do not seem to be included or integrated into the discussion

Response: In the discussion we have stated that the quantitative and qualitative results were in agreement with minimal discordance as shown on page 14, line 443-444. Where appropriate the agreement between the two methods has been emphasized in the discussion

22. What would common arrangements (insurance or free care) look like? How would they be financed? What about costing and pricing and cost-recovery, quality of care/standards – would governments be able to “sell” this to their citizens

Response: We meant that if there was health insurance across all East Africa nations with portable benefits for access, then that would be a form of common arrangement. We have highlighted this example in the discussion section on page 15, line 473-474

23. ‘Some border residents also preferred to go where care was of acceptable quality’ what did they not like about the quality of services in their place of residence? The statement about HIV seems to relate more to fear about being stigmatised or shame

Response: We have rephrased this statement to show that acceptability of care was viewed in terms of confidentiality and perceived quality of the services across the border on page 16, line 479-480

24. ‘From our study, we see the need to move from nationally circumscribed rights and entitlements to a legal environment that allows border residents to freely access care cross the border if the nearest service delivery point is on the opposite side of the border’ – so do you mean people are illegally crossing the border for HC?

Surely this is not new – what are government positions on this?

Response: We have added a phrase to clarify that border residents should access care within a well-established and agreed upon arrangement between countries as shown on page 16, line 487-488

25. ‘For UHC to be efficiently delivered, countries may need to revise their planning, resourcing and regulatory arrangements to enable border communities access the existing healthcare provision without bearing the hardship of navigating these barriers on their own. This can also reduce the uncertainty and discretion of front-line officials if they were all aware of the access rights for border residents in neighboring communities’ – not sure you have presented evidence to convince the reader this is an issue

Response: We have revised the statement to indicate the thought behind the suggestions at an analytical level as shown on page 16, line 490

Reviewer: 3

Dr. Hiroshi Aiura, Nanzan University

Comments to the Author:

The report on "Experiences of seeking health care across the border: Lessons to inform upstream policies and system developments on cross border health in East Africa" (bmjopen-2020-045575).

Summary

This paper explores the factors to access to health care services across the border, by a cross-sectional survey among 279 household adults. The linear regression analysis shows that the main predictors for ease of access for health care were "ease of border crossing", "small self-foundation", "Services and medicines availability" and "acceptable quality of services". The deductive content analysis shows the key facilitators for successful navigation of access barriers such as the presence of informal routes, speaking a similar language.

Comments

1. The results the abstract described have not been in the body of the paper. Especially, the abstract shows that nearly one third (77/279) of the respondents had accessed care from the opposite side of the border, but I could not find this result in the body of the paper. Therefore, I did not know whether the results of the paper are reliable.

Response: The statement has been corrected to show that 77/279 had accessed care more than a year ago and 64/279 less than a month ago as shown in the abstract in line 35-36 and the results in line 265-268 and in table 2, variable 2

2. I could not find (policy) implications based on the results of the linear regression analysis. I feel that the results of the linear regression analysis cannot sufficiently resolve the issues of this study.

Response: The quantitative and qualitative results were complementary in highlighting the factors (ease of crossing and affordability) that enabled access to care across the border. We have highlighted the complementarities in the discussion in line 451-453 and line 460-463.

3. If the aim of the study is to find the reasons why border residents cross or fail to cross the border, I am afraid that the author(s) should compare the group of the residents crossing the border with the group the residents not crossing.

Response: The specific objective in the introduction section has been reworded to clarify that the approach was not comparative. This is shown on page 5, in line 118-119

4. I could not judge whether the labels of latent variables are appropriate in the factor analysis, because I do not know what the list of questionnaires is.

Response: The questionnaire has been provided as uploaded for the editors

5. The paper did not show P values for each latent variable. If P values were high, latent variables would not statistically influence ease of access for health care.

Response: In Table 2, we have bolded the latent variables that were significant and added the P-values for each variable

6. "figure 1" in "Overall ease of cross border access to healthcare" at page 8 would be a typo of "figure 3".

Response: This has been corrected to "figure 3"

Reviewer: 1

Competing interests of Reviewer: None declared

Reviewer: 2

Competing interests of Reviewer:

Reviewer: 3

Competing interests of Reviewer: None.

VERSION 2 – REVIEW

REVIEWER	Seddighi, Hamed University of Social Welfare and Rehabilitation Science
REVIEW RETURNED	15-Oct-2021

GENERAL COMMENTS	The authors seem to have responded to the comments I made earlier. Therefore, in my opinion, this text can be published in its current form.
--

REVIEWER	Aiura, Hiroshi Nanzan University, Faculty of Economics
REVIEW RETURNED	04-Oct-2021

GENERAL COMMENTS	The author(s) has rewritten well according to my comments. I agree to publish this article.
---